# 1-PAGER: One Pass Answer Generation and Evidence Retrieval

**Palak Jain**[1]       **Livio Baldini Soares** [2]       **Tom Kwiatkowski**[2]
[1] Google Research       [2] Google Deepmind
{palakj,liviobs,tomkwiat}@google.com

## Abstract

We present 1-PAGER the first system that answers a question and retrieves evidence using a single Transformer-based model and decoding process. 1-PAGER incrementally partitions the retrieval corpus using *constrained decoding* to select a document and answer string, and we show that this is competitive with comparable retrieve-and-read alternatives according to both retrieval and answer accuracy metrics. 1-PAGER also outperforms the equivalent 'closed-book' question answering model, by grounding predictions in an evidence corpus. While 1-PAGER is not yet on-par with more expensive systems that read many more documents before generating an answer, we argue that it provides an important step toward attributed generation by folding retrieval into the sequence-to-sequence paradigm that is currently dominant in NLP. We also show that the *search paths* used to partition the corpus are easy to read and understand, paving a way forward for interpretable neural retrieval.

## 1 Introduction

In recent times, there has been a push to reformulate a wide variety of tasks from NLP and other domains into the sequence-to-sequence paradigm, to make use of large pre-trained Transformer networks (Vaswani et al., 2017). However, despite evidence that large language models can often answer questions (Roberts et al., 2020), predict identifiers of documents that support those answers (Tay et al., 2022), or generate text that contains and explains an answer (Yu et al., 2022) the dominant paradigm in question answering is still the retrieve-and-read approach that pipelines separate retrieval and answer generation modules. This approach has the benefit that it can provide direct and targeted paragraph-level attribution for the generated answers (Bohnet et al., 2022). However, it also relies on a heterogenous mix of models that are hard to train in concert (Metzler et al., 2021).

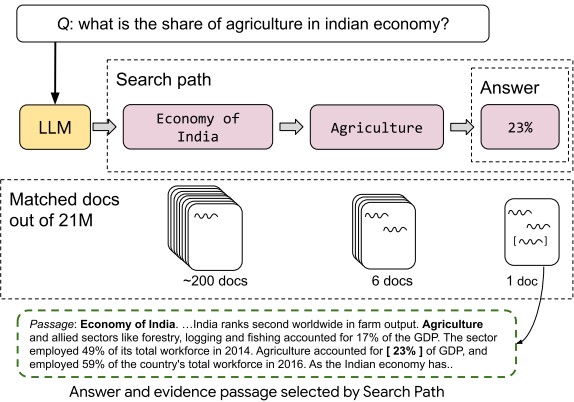

Figure 1: Example 1P output that iteratively partitions the corpus into sub-sets containing the generated n-grams. The last n-gram is taken as the answer.

Motivated by the observation that language model decoders already perform search over possible sequences (Graves, 2012), and that evidence documents themselves are simply sequences of tokens, we present an alternative approach that relies on a single Transformer model. In this approach, which we name 1-PAGER (One Pass Answer Generation and Evidence Retrieval) or simply 1P, the decoder iteratively partitions a corpus of evidence documents by generating a *search path* consisting of a set of keywords that identify relevant documents and an answer string that is contained in at least one of these documents. With 1P, we aim to explore the spectrum between CBQA, where the answer is generated without reference to an evidence corpus, and pipelined approaches that feed retrieved documents into the transformer.

Figure 1 illustrates an example in which the corpus is iteratively partitioned into documents that contain the string 'Economy of India', then those that also contain the string 'Agriculture', and finally those that also contain the answer string '23%'.

1P output sequences are guaranteed to match at least one document in the evidence corpus. This is enforced via a constrained decoder that has ac-

cess to an FM-index representation of the evidence corpus contents (Ferragina and Manzini, 2000) and we evaluate 1P's ability to correctly answer open-domain questions while also retrieving passages that provide support for those answers (Bohnet et al., 2022). Since 1P is the first model that can do both of these tasks, we compare to pipelined systems that first retrieve a single passage and then generate an answer based on this evidence passage. 1P is competitive as a passage retriever, performing similarly to a widely used dense retriever (Karpukhin et al., 2020) and outperforming the SEAL system which independently generates keywords rather than a search path (Bevilacqua et al., 2022). 1P also outperforms an equivalent closed-book question answering (CBQA) model (Roberts et al., 2020) according to answer accuracy. Part of this improvement comes from the prediction of search paths themselves, reminiscent of chain-of-thought reasoning (Wei et al., 2022), and part is from 1P's constrained decoder, which forces the model to generate answers from passages that contain the keywords.

While 1P does not yet perform as well as the very best retrieval or open-domain question answering systems in terms of accuracy, the fact that it is competitive with pipelined systems that are trained with the same data and which use similar amounts of inference-time compute suggests a promising path ahead. Unlike those systems, 1P can be trained end-to-end along with any other task that fits into the sequence-to-sequence paradigm. Additionally, 1P search paths are inherently interpretable, unlike embeddings used in dense retrieval.

## 2 Related Work

**"Retrieve-and-read" Question Answering** Question answering approaches in NLP are dominated by the "retrieve-and-read" paradigm where a retriever first fetches hundreds of relevant documents from a corpus, followed by a language model that reranks and extracts the answer (Harabagiu et al., 2003; Chen et al., 2017; Zhu et al., 2021). Sparse retrievers such as BM25 (Robertson et al., 2009) build a high-dimensional lexical index over text corpus. Dense retrievers (Karpukhin et al., 2020) use a dual encoder architecture to embed the query and document and perform an approximate nearest neighbor search. Various modifications to dense retrieval have been proposed over the years includ-

ing hard negative training (Xiong et al., 2020), late interaction (Khattab and Zaharia, 2020; Santhanam et al., 2022), few-shot learning (Izacard et al., 2022), joint retriever and reader training (Jiang et al., 2022).

A particular variant of interest is the **Iterative Retrieval** process where the query is reformulated incrementally (Das et al., 2019; Lee et al., 2022) leading to an interactive search process (Jiang et al., 2023; Adolphs et al., 2021). This query augmentation scheme has similarities with our use of search paths. However, we use the paths to iteratively partition the corpus while prior works have used it for refining the query.

To perform well, retrieve-and-read systems will typically retrieve 10s to 100s of passages that must be processed by a language model. In constrast, 1P retrieves and extracts an answer in a single pass of language model generation.

**Closed Book Question Answering** With data and parameter scale, LLMs in a closed-book setting (CBQA) have shown competitive performance (OpenAI, 2023; Anil et al., 2023; Yu et al., 2023) to retrieval pipelines (ODQA), however without producing any attributed passages (Rashkin et al., 2021; Bohnet et al., 2022). An extension of CBQA is post-hoc retrieval where a large language model LLM) is first used to generate an answer and then evidence for the question-answer pair is fetched by a retriever (Gao et al., 2023a; Bohnet et al., 2022). While post-hoc retrieval serves the same goal as 1P, it still uses a pipeline of LLM and retriever to do so.

**Generative Retrieval** Recently, generative retrieval has emerged as an alternative to the conventional "retrieve-and-read" pipeline (Metzler et al., 2021). Genre (De Cao et al., 2021) performed generative entity linking by constraining model's decoding to a set of entities. DSI (Tay et al., 2022) showed one of the first proof of LLM's ability to memorize docids in the corpus. However, atomic ids or hierarchical clusters, as used in DSI, are opaque identifiers and capture limited information. Works such as SEAL (Bevilacqua et al., 2022) and Ultron (Zhou et al., 2022) use a semantically richer representation: keywords in the document. In particular, SEAL constrains the generation to only keywords in the corpus using the FM-index (Ferragina and Manzini, 2000), a key data structure we borrow in this work.

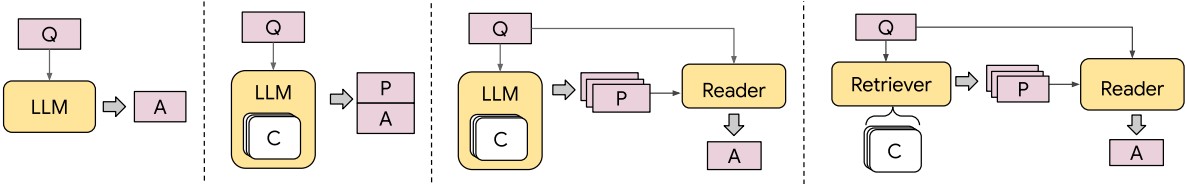

Figure 2: System illustration of different QA systems. From left to right: CBQA, 1-PAGER, SEAL, Retrieve-and-Read system. *C* denotes the retrieval corpus, *P* a retrieved passage, *Q* the input question and *A*, the generated answer. 1P is closest to CBQA (only single model used) but it also outputs a passage retrieved from *C*.

1P represents docids as keyword paths, which are arguably more interpretable, and learns a soft partition over the corpus instead of the hard partition imposed by DSI's clustering.

Another crucial distinction is 1P's ability to both retrieve and generate an answer while prior works rely on a external re-ranker/reader for the same. A high-level view of various question-answering systems is presented in Figure 2.

**Attributed Question Answering** Standard metrics for open-domain question answering, such as exact match or token-based F1, have received criticism for being imprecise and/or insufficient. Several efforts have proposed augmenting answers with textual evidence, via retrieval or citations (Bohnet et al., 2022; Menick et al., 2022; Gao et al., 2023b). While this work does not directly evaluate the quality of retrieved answer evidence, our proposed model inherently produces a passage to support the final answer, along with a search path of keywords, which could be used to provide users with answer evidence.

## 3  Iterative Corpus Partitioning and Answer Prediction

We focus on the problem of learning a mapping $f(q, D) \rightarrow (a, d_a)$ from a question $q$ and corpus of documents $D$ to an answer and supporting document $(a, d_a)$. The predicted document $d_a$ is retrieved from $D$ and the answer $a$ is a sub-string of $d_a$. The document $d_a$ should be relevant to the question and provide evidence for answer.

The goal of this paper is to model the function $f$ using a single sequence-to-sequence model, rather than a pipeline which first retrieves $d_a$ and then feeds it into an answer generation module. To achieve our goal, we recast retrieval as an *iterative corpus partitioning* process illustrated in Figure 3.

**Iterative corpus partitioning** adopts the LM decoder's autoregressive search process to partition

$D$ by predicting n-gram *keywords*.

An n-gram of tokens $k$ is said to be contained in a document $d$, denoted by $k \prec d$, when $k$ is a sub-sequence of $d$. We define a *keyword* corpus partitioning function

$$\mathcal{F}(D, k) = \{d | k \prec d; d \in D\}$$

that selects only those documents that contain $k$. 1-PAGER iteratively partitions the corpus $D$ by generating a sequence of n-grams that we refer to as a *Search Path* $p_t = [k_1, k_2, \ldots, k_t]$. Each prefix of this search path defines a subset of $D$ via the *search path* corpus partitioning function

$$\mathcal{P}(D, p_t) = D_{p_t} = \{\cap_{i \in [1,t]} \mathcal{F}(D, k_i)\}$$

and each subsequent keyword $k_{t+1}$ narrows down $D_{p_t}$ into further sub-spaces such that $D_{p_{t+1}} \subseteq D_{p_t}$.

**Answer prediction** is treated in exactly the same way as keyword selection and in 1P the last keyword from $p$ is taken as the answer.

## 4  Constrained Decoding and FM-Index

To avoid generating empty partitions, we constrain 1-PAGER to only decode search paths that match at least one document. We modify the decoder's beam-search strategy to only allow keyword continuations that are contained in the current partition.

Given a document subset $D_{p_i}$, which could be the full corpus $D$ at the start of decoding ($i = 0$), and a keyword prefix $k$, which could be empty, the set of all valid continuation tokens is defined as,

$$\mathcal{C}(k, D_{p_i}) = \{x | \; k \| x \prec d, d \in D_{p_i}\}$$

where $x$ is any vocabulary token and $\|$ indicates concatenation of two token sequences. As a special case, when $k = \phi$ and $i = 0$, all tokens in $D$ are valid continuations. 1P separates keywords in $p_T$ with a special separator token $\rightarrow$ and marks the end of the sequence with an EOS token. These two tokens are always valid continuations.

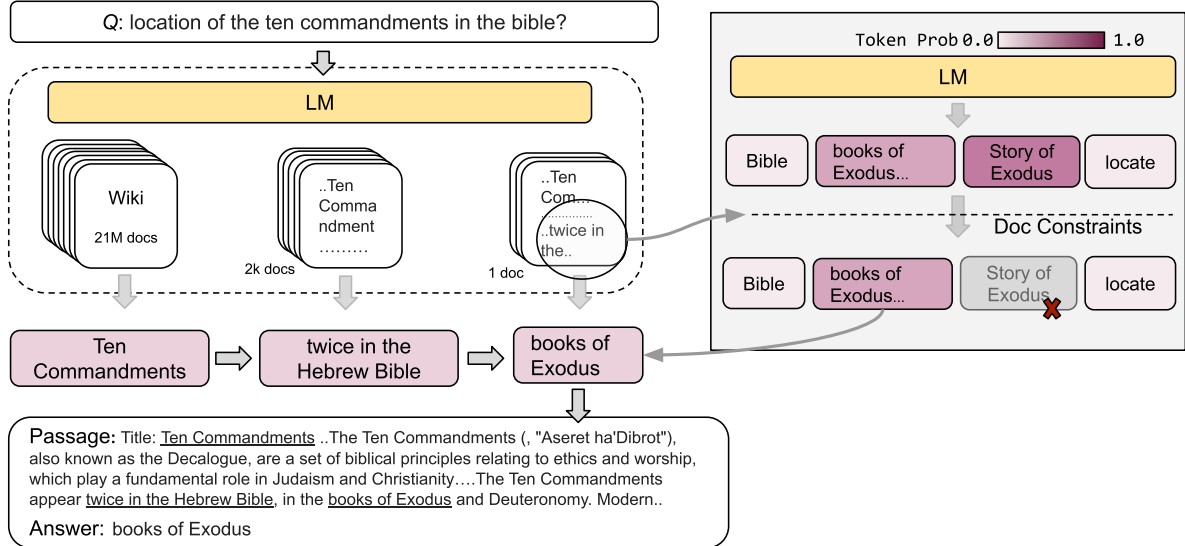

Figure 3: Illustration of the 1P decoding process. A keyword can only be generated from the documents matching previously generated keywords. Right panel shows a magnified view of applying constraints to a decoding step. Any keyword not present in the documents is masked out.

Consider Figure 3. The three keywords correspond to the decoded token sequence [*Ten, Commandments, →, twice, in, the, Hebrew, Bible, →, books, of, Exodus,* EOS]. At the start of decoding, any token in $D$ is allowed. After decoding *Ten*, only those tokens that follow *Ten* as an n-gram in $D$ are allowed, along with the default separators. After decoding [*Ten, Commandments, →*] we are ready to start a new keyword, but only tokens from documents that contain the keyword *Ten Commandments* are allowed. Decoding continues in this manner until EOS is generated.

To efficiently implement these constraints, we need a data-structure that can quickly determine both $\mathcal{C}(k, D_p)$, the continuation tokens given a document set and $\mathcal{P}(D_p, k)$, the subset of documents that contain a given path.

For this, we extend the usage of an FM-index (Ferragina and Manzini, 2000) as described in (Bevilacqua et al., 2022). The FM-index is a compressed token-based index over a corpus $D_0$ with a few important properties for our usage: (1) it can efficiently list possible token continuations for a sequence prefix that occur in $D_0$ i.e., $\mathcal{C}(k, D_0)$, (2) it can list the set of documents in the corpus that match an n-gram i.e., $\mathcal{F}(D_0, k)$, and (3) it supports search over arbitrary n-grams that occur *within* documents. Note that the FM-index operations are optimized for $D_0$, the original corpus it is built over. We extend these to an arbitrary $D_p \subset D_0$ at additional cost described in Appendix A.1.

## 5 Training data generation

For training 1P, we produce a dataset with examples of queries and search paths as described above. At a high-level, we generate search paths by iteratively selecting n-grams from an answer passage, and simulating, using the FM-Index of the retrieval corpus, the partitioning of the corpus after selecting each keyword, until only a few documents remain. Finally, the answer span $a$ is appended to the search path. Each example produced can be serialized as sequence-to-sequence pair of inputs and targets as:

```
inputs: Generate keywords for: <q>?
targets: K_SEP k_0 K_SEP k_1 ... K_SEP A_SEP a EOS
```

### 5.1 Keyword Selection

A good keyword should have a) high relevance to the query and b) effectively narrow down the search space. To identify relevant keywords, we restrict to only the gold document $g$. All ngrams in $g$ of length up to five are extracted. Irrelevant keywords are filtered out such as those starting or ending with stop words. Similarly, keywords that are too rare in the corpus, e.g., "Philippines at Luzon" or too frequent, e.g., "part" are excluded based on a threshold on their count in corpus. The remaining keywords are scored with a combinations of heuristics, mainly Rouge-1 similarity with the query (Lin, 2004) along with minor award for keywords containing entities and penalty for keywords highly frequent in the corpus.

This scoring mechanism often misses out on keywords that are semantically relevant, but do not lexically overlap with the query. To boost the relevance of our keyword set, we re-score the top hundred keywords using a language model. A T5-XXL model is finetuned with the input as the query $q$ and target as either the title or a heuristically sampled keyword in a similar fashion to Bevilacqua et al. (2022). The heuristically sampled keywords are re-scored using this model to obtain a refined LM-scored set. Two other special types of keywords are awarded high scores: Title of the gold passage and the keyword containing the answer string $a$.

## 5.2 Search Paths

The first keyword in a search path needs to effectively partition the corpus. We experiment with either the title or the highest scored keyword from the gold passage as the first keyword in the path. The next keywords are sampled based on their score, given they do not overlap with any of the existing keywords in the path. We continue augmenting a path $p$ with keywords until at most ten passages in the corpus match i.e., $|D_p| < 10$. The answer keyword is then appended to the path. Our train paths (including the answer) contain a median of three keywords and one matching document.

## 6 Experimental Setup

### 6.1 Datasets

We use Open-NQ (Kwiatkowski et al., 2019; Lee et al., 2019) as the question-answering dataset for training. For evaluation, besides Open-NQ, WebQuestions (Berant et al., 2013) and CuratedTREC (Baudiš and Šedivỳ, 2015) are used to measure out-of-domain performance. The FM-Index corpus for constrained decoding is built over DPR Wikipedia corpus with 100-word splits (Karpukhin et al., 2020). The positive gold passages from DPR are used for sampling training paths. This setup is chosen to mirror SEAL and also permits fair comparison against DPR.

### 6.2 Training

1P's training dataset contains 310k paths corresponding to 55k queries from Open-NQ. Majority of the training paths begin with the title, with a small fraction starting with other keywords (12%). All keywords, except the title, are scored using the LM-scoring technique described above.

For our experiments, we use the T5X (Roberts et al., 2022) framework. A T5-XXL 1.1[1] (Raffel et al., 2020) model is finetuned with a batch size of 256 and dropout of 0.1. No additional hyperparameter tuning is performed. We format search paths using the reserved tokens K_SEP = "<extra_id_0>" and A_SEP = "<extra_id_1>".

### 6.3 Inference

Our best model employs beam decoding with a beam of 5. Even when the beam is greater than one, only the top-beam result is used for retrieval. We discuss the effect of beam size in depth in Section 7. Given the top generated path $p$, $D_p$ corresponds to the retrieved documents. In case $|D_p| > 1$, a document is sampled arbitrarily for evaluation.

### 6.4 Baselines

We compare to a closed-book question answering (CBQA) system that generates answers, but does not ground these in an evidence corpus, as well as retrieve-and-read systems that combine a variety of retrievers with a Transformer-based answerer module. Both the CBQA baseline and the answerer module are derived from the same T5-XXL 1.1 pretrained model as 1P.

#### 6.4.1 T5-CBQA

A T5-XXL 1.1 model is fine-tuned to predict answers from the DPR training set for 10,000 steps with a batch size of 128. Note that it is possible to achieve a higher closed-book performance on NQ using the full Open-NQ training split instead of the subset included in the DPR training set (Roberts et al., 2020). However, to enable meaningful comparison we restrict the CBQA baseline to the same training examples used to train 1P.

#### 6.4.2 Retrieve-and-Read

The retrieve-and-read baselines first retrieve a *single* passage from the evidence corpus, and then feed this passage and the question into the answer generation module[2]. We report retrieval accuracy for the retrieved passage and answer accuracy for the generated answer.

**T5-Reader** We tune a T5-XXL 1.1 model to generate answers from (question, evidence passage)

---

[1]https://goo.gle/t5-checkpoints
[2]This differs from ODQA evaluations that do not include evidence retrieval as a first-class task, where many retrieved passages are fed into a reader that generates an answer without attribution to any single piece of text.

pairs. This is the same base model used by 1P and we train on the (question, passage, answer) triples in the DPR training split to ensure fair comparison.

**DPR-Retriever** We compare against vanilla DPR finetuned on NQ without hard negatives (Karpukhin et al., 2020) using the pre-computed index available on DPR's repository[3]. We note that our ODQA setup differs from the one used by Karpukhin et al. in that we choose the highest scoring retrieval as evidence for answer generation, instead of generating from the top-100 passages without attribution.

**BM25-Retriever** We use Pyserini toolkit (Lin et al., 2021) with default configurations, retrieving the top-1 passage.

**SEAL-Retriever** SEAL (Bevilacqua et al., 2022) is a generative retrieval system that generates a set of keywords constrained on the corpus. In terms of technique, 1P borrows inspiration from SEAL's use of the FM-Index as well as keywords-as-identifiers. However, the two setups have substantial differences that we highlight in Section 8. We run SEAL with its default configuration and a beam of 5 using the publicly released checkpoint based on Bart-large (Lewis et al., 2020). All outputs from the beam are used for retrieval.

### 6.5 Evaluation

We evaluate in-domain performance on the Open-NQ test split and out-of-domain performance on WebQuestions (WQ) and CuratedTREC (TREC) following the setup from Karpukhin et al. (2020). Passage retrieval performance is measured with Hits@1 using Pyserini evaluation scripts[4].

### 6.6 1P configurations

We experiment with three configurations: a) 1P: Our primary setup that uses both training and constrained decoding procedures described above, producing a retrieved passage as well as an answer. b) 1P-Unconstrained: Only the training technique described in Section 5 is adopted, with standard unconstrained decoding. Since generation is unconstrained, it is possible that no passage gets retrieved for a given path. c) 1P + Reader: Here, we take the top retrieved passage from 1P and input it to the Reader model (Section 6.4) to extract the answer.

---

[3] https://github.com/facebookresearch/DPR
[4] https://github.com/castorini/pyserini

## 7 Results

| Retriever | Answerer | Retrieval Hits @1 | Answer EM | Answer F1 |
|---|---|---|---|---|
| – | T5 - CBQA | – | 26.8 | 34.0 |
| BM25 | T5 - Reader | 23.6 | 17.9 | 24.0 |
| SEAL | T5 - Reader | 37.9 | 29.4 | 35.8 |
| DPR | T5 - Reader | 46.5 | 35.6 | 42.4 |
| 1P | T5 - Reader | 46.3 | 34.2 | 41.4 |
| 1P - Unconstrained | | 29.3 | 29.3 | 36.1 |
| 1P | | 46.3 | 31.7 | 38.0 |

Table 1: Comparison of different Retriever and Answerer combinations on the NQ-Open test set. In retrieve-and-read setups, answers are generated from the top-1 retrieved passage. 1P combines passage retrieval and answer generation in a single prediction.

| System | WebQuestions Hits @1 | WebQuestions EM | TREC Hits @1 | TREC EM |
|---|---|---|---|---|
| BM25 + Rdr | 19.7 | 14.2 | 35.2 | 29.1 |
| DPR + Rdr | 32.0 | 17.3 | 51.6 | 35.0 |
| 1P + Rdr | 38.0 | 20.4 | 63.8 | 38.5 |
| 1P | 38.0 | 20.5 | 63.8 | 36.4 |

Table 2: Comparison of different Retriever and Answerer combinations on Out-of-domain datasets. Both the Retriever and Answerer (Rdr) are trained on only Open-NQ. In retrieve-and-read setups, answers are generated from the top-1 retrieved passage.

We compare to the baselines described in Section 6.4 on Open-NQ using both retrieval and answer accuracy metrics in Table 1. Answers are generated based on the top retrieved document in systems that separate retrieval from answer generation, to provide a clean comparison between systems that return (answer, evidence passage) pairs. Table 2 reports the out-of-domain performance of various systems on WQ and TREC.

1P outperforms CBQA in question answering and beats the retrieve-and-read systems, BM25 and SEAL. On the passage retrieval task, it significantly improves over BM25 and SEAL. For in-domain setting, 1P is competitive with DPR on retrieval task, but lags behind the QA pipeline that uses DPR. However, this appears to be more due to the reader rather than the retriever as discussed in Section 8. It is worth noting that 1P general-

izes significantly better out-of-domain compared to other systems.

**Utility of Search Paths**  1P-Unconstrained can be viewed as an extended version of CBQA that generates a search path before predicting the answer. Thus, improvement of 1P-Unconstrained over CBQA can be attributed to this path-conditioned answer generation process, analogous to chain-of-thought reasoning (Wei et al., 2022; Lampinen et al., 2022).

| System | Constrained Decoding | Beam | |
|---|---|---|---|
| | | 1 | 5 |
| CBQA | No | 26.7 | 26.8 |
| 1P Unconst. | No | 29.0 | 29.3 |
| SEAL + Reader | Yes | 28.5 | 29.4 |
| 1P | Yes | 28.7 | 31.7 |

Table 3: EM for various decoding setups with different beam sizes on Open-NQ. Only top-beam result is used for evaluation, except in SEAL which uses all beam outputs. 1P constrained decoding benefits the most from a large beam whereas Unconstrained setups have only a slight effect.

**Effect of Constrained Decoding**  The purpose of constrained decoding is to ground the answer in an evidence retrieved from the corpus. As expected, the constrained setup enables 1P to achieve a higher Hits@1 than 1P-unconstrained. Surprisingly, when decoding with a beam of one, we observe a small drop in answer accuracy for 1P compared to 1P-Unconstrained (Table 3). Inspecting the losses, two dominant reasons surface. Firstly, As DPR passages are chunked into 100-words (Karpukhin et al., 2020), some queries may become unanswerable given a single passage due to missing context. This is disadvantageous when the model has memorized the answer but there is no single passage to attribute it to.

Secondly, during constrained decoding, after generating the initial keywords, the search space may soon become sparse with no good candidates to pick from. Could a larger room for planning its actions help the model here? Indeed, increasing the beam size to 5 improves performance by 3% (Table 3), even when only the top-beam is used for retrieval. We refer to this as **Planning**, since the larger beam only enables the model to plan better and the remaining beam outputs are otherwise dis-

carded. Note that unconstrained decoding does not gain from planning. In the final setup in Table 1, we use a beam of 5 for both 1P and SEAL. Unlike 1P, SEAL uses all the outputs from the larger beam for retrieval.

## 8  Discussion and Ablations

**Generating Answers**  While 1P is capable of generating answers, Table 1 highlights that it falls behind the 1P+Reader. The reason seems to be clear: the Reader has visibility into the full passage context while 1P is limited to the decoded search path and the constrained index which only ensures that generations are grounded in the corpus. Since 1P does retrieve passages, it would be possible to pull in the corresponding text as input for answer generation. We leave this as future work.

**Comparison to SEAL**  While 1P takes inspiration from SEAL, in practice, there are a few key differences between the two systems aside from 1P's answer generation.

SEAL generates a large set of keywords (Table 4) using many separate decodes and heuristic guidance (Appendix A.3). In contrast, 1P decodes a single sequence of about three keywords.

| | SEAL | 1P |
|---|---|---|
| Median keywords | 32 | 3 |
| Median docs retrieved | 500 | 1 |
| Generates answer | ✗ | ✓ |

Table 4: Key differences between SEAL and 1P measured over Open-NQ test split with a beam of 1.

The SEAL keywords are a set, decoded independently of each other and re-scored using sophisticated techniques to retrieve a large number of documents. For instance, the default configuration in SEAL retrieves up to 500 documents. This makes SEAL suitable to be employed in conjunction with a re-ranker. In contrast, 1P search path's map directly to a single (or few) relevant documents (Appendix A.6).

We acknowledge the model-size variation between SEAL and 1P in the reported experiments, however we preferred using the publicly available SEAL checkpoint. Given the discrepancies with larger beam-size, multiple decodes and use of Reader model, it is difficult to have an apples to apples comparison between the two systems.

**Path vs Keyword set** We qualitatively observe that keywords in a 1P path, owing to sequential generation, are distinct and add new information as compared to the SEAL output set where overlapping keywords are common (Appendix A.3). Thus, paths are advantageous for precisely narrowing down to a single relevant document while keyword sets are effective for retrieving a large number of documents that can later be reranked. This is corroborated by the fact that 1P is better at Hits@1 while SEAL is better at Hits@5 (Appendix A.4).

**Qualitative Analysis** Table 5 illustrates patterns of Search Paths generated by 1P. We note some of the common path patterns here:

1) First keywords are entities in the query, followed by query predicates that iteratively narrow down towards an answer. This is the most common type of path observed and can be attributed to the dominant presence of title in the training data.

2) Rewrites of the original query or related predicates such as "seasons consists of", "appeared on ...". Such paths are more prevalent where there is no canonical entity in the query or no entity can be determined with high confidence.

3) Answer is directly generated followed by supporting keywords that guide towards an attributed passage. This happens in a small fraction of cases, likely where the pretrained model has memorized an answer with high confidence.

Overall, we find the generated search paths to be fairly meaningful and interpretable.

**Sampling Search Paths for Training** Table 6 highlights that high quality keywords are crucial to performance. The LM re-scored set of keywords result in significant accuracy gain over heuristically sampled keywords. Paths with first keyword as Title boost performance further. Mixing in a small fraction of paths starting with non-title keywords encourages the model to generate predicates where no entity can be determined, giving us the best results.

**Sensitivity to tokenization** We find that constrained decoding is highly sensitive to rare tokenization or punctuation formatting in the corpus. Consider the query "who sang i ran all the way home" with the gold document title "Sorry (I Ran All the Way Home)". In the unconstrained setup, the model's top prediction starts with "I Ran All the Way Home". However, "(I" is tokenized differently from "I" and searching over the FM-Index

returns no match. As a result, constrained decoding drops the predicted keyword altogether, resorting to lower ranked keywords in the beam. We partially fix the issue by modifying the answer in a fraction of the training data to include surrounding punctuation tokens based on how they appear in the FM-index. For instance, the keyword "I Ran ..." would update to "(I Ran ...". This simple change leads to a jump in answer accuracy from $26.4\%$ to $28.7\%$. However, much more work is needed to make 1P robust to variations in tokenization.

See Appendix A.2 for analysis of training data size and Appendix A.5 for masking logits vs logprobs.

## Conclusion

We introduce 1-PAGER, the first system to perform question answering and passage retrieval in one pass with a single language model, using a constrained decoder to iteratively partition the retrieval corpus and then generate an answer. We show competitive or improved performance over a variety of comparable baselines and carefully analyze the results, ablating both training strategies and decoding style. We also provide a qualitative analysis of predictions to illustrate the system's capabilities. Challenges with constrained decoding are surfaced including poor search spaces and sensitivity to tokenization and mitigation strategies are presented.

We hope that 1P adds value in demonstrating how a single transformer model can be harnessed to do both retrieval and answering and pave the path for further progress in the generative retrieval domain.

## Limitations

1P is geared towards identifying a concise, small set of documents and generating answer in a single go. While this makes the architecture simpler, it also adds certain weaknesses. 1P is not effective for retrieving a large number of documents and falls behind pipelined systems that combine retrieval with re-ranking. Even for a single passage, it lags behind state-of-the-art dense-retrieval techniques. 1P's method of answer generation is also not competitive with the use of a reader, due to lack of passage context.

Our training strategy relies heavily on titles or entities and it's generalization on corpora without rich structure or on queries without central entities, remains to be studied.

| Query (Q) and Generated Search Path (SP) | | Comment |
|---|---|---|
| *Correctly attributed passages and answers* | | |
| Q: how many episodes of greys anatomy season 14
SP: Grey's Anatomy (season 14) » season consists of 24 episodes » 24 | | Query entity resolved first,
followed by query predicates |
| Q: when did they start adding zinc to pennies
SP: Penny (United States coin) » zinc » Lincoln cent » 1943 | | Query entity resolved
iteratively |
| Q: who was executed for being an american spy during the revolutionary war
SP: Nathan Hale » Army during the American Revolutionary » Nathan Hale | | Answer generated first |
| Q: who was the grandfather on the cosby show
SP: appeared on "The Cosby » Earle Hyman | | Query rewrites |
| *Incorrect Passage or Answer* | | |
| Q: who decides the number of judges in the high court
SP: judge is appointed » High Court » Chief Justice of India | A: President of India | Path correctly resolved,
Failed on answer |
| Q: when did the isle of wight become an island
Isle of Wight » 1890 » 1890 | A: During the last Ice Age | Query entity resolved,
Failed on supporting keywords |
| Q: love yourself by justin bieber is about who
SP: Love Yourself: Her » music video » Her | A: Rihana | Failed to resolve
query entity |

Table 5: Example 1P Search Paths (SP) on Open-NQ test set. The last keyword in SP is the predicted answer. Gold answers are indicated by A.

| Search Path | Hits@1 | EM |
|---|---|---|
| Heuristic | 34.5 | 22.6 |
| LM-scored | 40.0 | 27.2 |
| Title » LM-scored | 41.9 | 28.0 |
| Title » LM-scored +
LM-scored (7+1) | 42.9 | 28.7 |

Table 6: Comparison of Training Search Paths on Open-NQ. Here LM-scored denotes re-scoring by LM on a heuristic set. All results are with a beam of one. "»" indicates keyword separator and "+" mixture of path types in the give ratio.

Constrained decoding also comes with its own challenges. Constrained beam outputs often lack diversity, so that even with a larger beam one may still end up in poor search spaces. Computing document-level constraints across the corpus is expensive as it may require scanning a large number of rows in the index. Further, communication between FM-Index and Transformer model slows down inference.

## Acknowledgement

We thank Don Metzler, Nicholas FitzGerald, Partha Talukdar, Srini Narayanan, as well as our anonymous reviewers, for their thoughful comments and valuable feedback

## Ethical Considerations

While Large Language Models can solve a wide range of tasks effectively, they also suffer from biases across axis such as gender, race, region (Chan, 2023). LLMs are also prone to generating toxic content, especially when probed about it. Although, our task grounds the model's generations on a corpus, some of the biases in pre-trained LLMs, may seep in 1-PAGER.

Building the FM-index and constrained decoding is a compute-intensive affair. We have experimented over a single dataset, Natural Questions, involving only knowledge-seeking queries, and single model family, T5. It is possible that some of our findings may not hold over other datasets or model families. Finally, our experiments are limited to English corpus and queries. The proposed approaches are resource-intensive and may not be accessible or valid for several low-resourced languages.

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

# A Appendix

## A.1 Constrain Computation

1P relies on two key operations for constrain computation:

a) $\mathcal{F}(D, k)$ : Documents that contain keyword $k$

b) $\mathcal{C}(k, D)$ : Next tokens for keyword $k$ in arbitrary document set $D$

$\mathcal{F}(D, k)$ is preprocessed and cached to allow for quick computation. $\mathcal{C}(k, D)$ is trickier to compute. When D represents the full corpus, FM-index can fetch the next tokens in $O(|V|log(|V|))$, where $V$ is the token vocabulary and independent of $|D|$. However, arbitrary $D$ requires a traversal over all documents and can be very expensive. In practise, the LLM training guides it to generate effective keywords such that $|D|$ is small.

We also apply certain other optimizations to reduce the compute cost:

- Constrains are computed lazily over a decoding pass.

- Several computations are cached, eg: keyword to document id mapping

- To cap the cost of constraints at each decoding step, we allow for unconstrained generation in rare scenarios, when the estimated cost is too high. If the generated path is absent in the corpus (<1% examples), these can be filtered out later.

Despite these optimizations, inference continues to be expensive and we perhaps need a special data structure for next token look-up.

## A.2 Training data size

| Dataset | Queries | Paths | Hits@1 | EM |
|---|---|---|---|---|
| Open-NQ | 55k | 55k | 41.9 | 28.1 |
| Open-NQ | 55k | 310k | 42.9 | 28.7 |
| Open-NQ + PAQ | 55k + 9M | 310k + 9M | 43.6 | 29.5 |

Table 7: Comparison of different dataset sizes for queries and paths

In Table 7, we observe the effect of dataset size on performance. Increasing the numbers of paths sampled per query improves performance, perhaps due to higher diversity in training. However, this method of dataset expansion is limited by the number of relevant paths we could extract for a query.

We also experiment with increasing the query set manifold by mixing in unsupervised datasets. A total of 9M QA pairs are sampled from PAQ (Lewis et al., 2021), a synthetic QA dataset, and search paths extracted with heuristic scoring described in Section 5. The original 1P training dataset is mixed in 1:1 ratio. This further boosts performance, but not proportionally to the amount of data added, indicating diminishing returns from silver datasets.

## A.3 SEAL keywords

SEAL generates a set of document substrings constrained on the corpus, that are combined to form document identifiers. Besides using a LM to generate keywords, SEAL utilizes several other mechanisms for extracting keywords. This includes partial beam sequences, heuristically adding query n-grams, sampling the top-k tokens from the logprobs of the first decoding step, force decoding title etc. The keywords are re-scored using the LM as well as FM-index count and all keyword combinations are retrieved. Table 8 illustrates keywords generated by both the systems. Note that SEAL keywords can be repetitive and therefore require a large number of keywords to narrow down to meaningful documents. This also makes SEAL suitable for retrieving a much larger set of documents that can be re-ranked later. The maximum number of retrieved documents for SEAL are capped by a hyperparameter with default value of 500. In contrast, 1P is geared towards retrieving only the top-document.

## A.4 Hits@5

SEAL does significantly better than 1P for Hits@5 (Table 9). We attribute this to the large set of keywords generated by SEAL as explained in the Appendix A.3.

## A.5 Normalizing sequence likelihood over constrained space

During constrained decoding a sequence $X$, we need to choose the next token from $\mathcal{C}(X, D)$ and not the entire vocabulary space $V$. Should the sequence likelihood be re-normalized over this constrained space? We find that re-normalizing the probabilities results in inflated likelihoods, making it hard for the model to back-track.

Consider the query, "where did the butchers in the slaughterhouse cases live" to which our model

| System | Question or Search Path | Answer |
|---|---|---|
| 1P | who has the most catches in nfl history | Jerry Rice |
|  | 2,000-yard club » Barry Sanders | Barry Sanders |
| SEAL |  Michael Irvin @@, yards per catch, caught his, touchdown, record | T.J. Houshmandzadeh |
| 1P | when was harry potter and the philosophers stone published | 1997 |
|  | Harry Potter and the Philosopher's Stone » first published in the United » 1997 | 1997 |
| SEAL |  Harry Potter and the Philosopher's Stone @@, "Harry Potter, Potter and the Philosopher's Stone is, Potter and the Philosopher's Stone Harry, novel | 1999 |
| 1P | what is the meaning of the harp in ireland | the arms of Ireland |
|  | Harp » national symbol of Ireland » national symbol of Ireland | national symbol of Ireland |
| SEAL |  Harp @@, Irish harp,, harp is, harp was, harp | aristocracy |
| 1P | who was the president of pakistan during 1971 war | Yahya Khan |
|  | Indo-Pakistani War of 1971 » Prime Minister of Pakistan » Zulfikar Ali Bhutto | Zulfikar Ali Bhutto |
| SEAL |  Indo-Pakistani War of 1971 @@, East Pakistan, Pakistani, Pakistan Army, Pakistan's | Muhammad Yaqub Khan |
| 1P | when do you declare honors in contract bridge | any time after the auction |
|  | Contract bridge » declaring » end of the hand | end of the hand |
| SEAL |  Contract bridge @@, declarer, bidding, honors, hands | bidding |

Table 8: Comparison of keywords generated by SEAL and 1P for randomly sampled exampled from Open-NQ test set. For 1P, we show the full search path separated by "»" with the last keyword as the answer. For SEAL, we illustrate the top-5 keywords along with the answer from Reader model. "" and "@@" are special tokens used by SEAL for identifying start of passage and title marker respectively. The Answer next to the question is the gold answer while others are predictions from corresponding systems.

| System | Beam | Hits@5 |
|---|---|---|
| SEAL | 1 | 59.7 |
| SEAL | 5 | 62.8 |
| 1P | 1 | 46.5 |
| 1P | 5 | 50.8 |

Table 9: Hits@5 on Open-NQ test. SEAL achieves a much higher score than 1P owning to the larger number of documents matched and re-scored. Note that only top-beam result is used for 1P while SEAL uses all beam outputs.

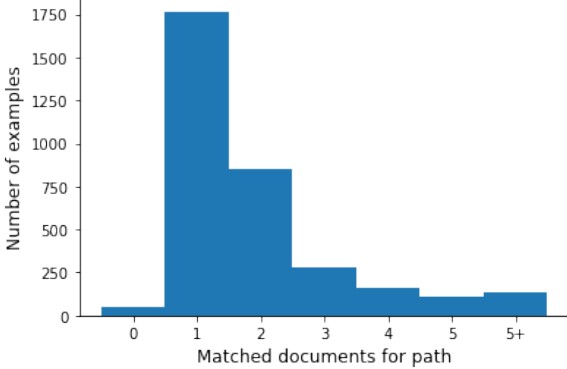

Figure 4: Number of matching documents in the corpus for 1P generated path in the test set. About half the examples match only a single path.

predicts an irrelevant search path [*Slaughterhouse Five*, *but*, EOS]. What's going on under the hood? The first keyword is incorrect lending the model into a poor search space. With the second keyword, the model is possibly looking to generate "butcher" but there's no such keyword in the constrained set. Ideally, the model should backtrack at this point to other candidates in the beam. However, since the set of continuations is small, renormalizing inflates the probablities of all tokens in $\mathcal{C}$ including *EoS*, even though the true likelihood of such a sequence is very low. Indeed, using the language model's scores directly without any re-

normalization cures this issue yielding [*Slaughterhouse cases*, *Butcher*, EOS]. and this is the strategy we opt for in all our experiments.

### A.6 Number of matching documents

1P generated paths effectively narrow down the corpus, generally matching only a few documents in the corpus as illustrated in Figure 4. Note that a small fraction of paths match 0 documents due to pruning optimizations applied during inference

time detailed in Appendix A.1.