# OpenReview forum: "1-PAGER: One Pass Answer Generation and Evidence Retrieval"
_EMNLP/2023/Conference — EMNLP 2023 Findings_

### Official Review · Reviewer_Sf64 · 2023-08-03

**Typos Grammar Style And Presentation Improvements:** 1. In the abstract, "While 1-PAGER is…
**Soundness:** 2

**Excitement:**

3: Ambivalent: It has merits (e.g., it reports state-of-the-art results, the idea is nice), but there are key weaknesses (e.g., it describes incremental work), and it can significantly benefit from another round of revision. However, I won't object to accepting it if my co-reviewers champion it.

**Paper Topic And Main Contributions:**

This paper proposes a single language model for both retrieval and answer generation for open-domain QA. Like previous work SEAL, the proposed method generate keywords from query to retrieve documents containing the generated keywords with the help of FM-index. Different from SEAL, this work generates a sequence of keywords to gradually narrow down to a subset of the corpus, and then directly generate the final answer after keywords generation. The experimental result shows the method performs worse than baselines.

**Questions For The Authors:**

1. What are the run time for inference? How is it compared to other method?
2. What are the exact solution for the constrained decoding with FM-index? The explanation in appendix is unclear and difficult to reproduce.


**Reasons To Accept:**

1. Generating a sequence of keywords to partition the corpus is a new technique in generative retrieval.
2. A single model for both passage retrieval and answer generation.

**Reasons To Reject:**

1. The decision to use a single model for both retrieval and answer generation seems forced and counter-productive. As shown in the experiment section, using a separate reader is superior to the proposed single model solution.
2. As pointed out by the paper itself, constrained decoding with FM-index is very expensive for keywords after the first one, which limits the value of the work in practice.
3. The comparison with SEAL is not fair, as the model of SEAL (bart-large) is much smaller than the one used in the current work (t5-xxl).

**Reproducibility:**

3: Could reproduce the results with some difficulty. The settings of parameters are underspecified or subjectively determined; the training/evaluation data are not widely available.

**Reviewer Confidence:**

1: Not my area, or paper was hard for me to understand. My evaluation is just an educated guess.

---

> ### Author Rebuttal · Authors · 2023-08-29
>
> Thank you for your careful reading and insightful critique. We have provided some responses to your feedback below:
>
> 1. > “The comparison with SEAL is not fair, as the model of SEAL (bart-large) is much smaller than the one used in the current work (t5-xxl).”
>
> We agree that the model size difference between SEAL and 1P is not desirable, however we preferred using the publicly available SEAL checkpoint (Bart-large) instead of training our own. Note that there are several other discrepancies between SEAL and 1P:
>
> a) *Inference time FLOPS*: SEAL uses significantly higher accelerator FLOPS than 1P. SEAL runs about 3 decoding passes. Besides, SEAL also re-scores all its keywords (30 on average), which is equivalent in cost to decoding. In contrast, 1P runs a single decoding pass of ~3 keywords.
>
> b) *Beam-size*: The reported results used a beam=5 for SEAL retrievals while 1P uses only top-beam for retrieval.
>
> c) *Heuristic keyword generation*: As discussed in Appendix A4, besides LM keyword generation, SEAL uses a variety of other heuristics such as extracting ngrams from the query, force decoding the title, fetching all high scoring next-tokens from the model given the query, partial beam matches etc.
>
> Given these discrepancies, it is difficult to have an apples-to-apples comparison.
>
> 2. > “The decision to use a single model for both retrieval and answer generation seems forced and counter-productive. As shown in the experiment section, using a separate reader is superior to the proposed single model solution”
>
> In this paper, we aim to explore the spectrum between CBQA, where the answer is generated without reference to an evidence corpus, and pipelined approaches that feed retrieved documents into the transformer.
>
> We agree that the pipelined approach is still most performant and do not yet argue for 1P as the best solution for anyone aiming to build the most accurate QA system possible. However, we believe that 1P's improvement over the equivalent CBQA model is significant and we also believe that 1P adds value in demonstrating how a single transformer model can be harnessed to do both retrieval and  answering, which has not been done before.
>
>
>
> 3. > “What are the exact solution for the constrained decoding with FM-index? The explanation in appendix is unclear and difficult to reproduce.”
>
> Thank you for flagging this, we have presented a summary below and would add a more detailed explanation to the Appendix. 1P’s Constrain decoding has 3 key functions to compute:
>
> a) Next tokens following the keyword k
>    - How: FM-index contains auxiliary data structures to compute this efficiently.
>    -  Time: O(Vocab)
>
> b) Documents that contain keyword k
>    -  How: All occurrences of k can be located by the FM-index in constant time. We iterate through each occurance and a hashmap provides the corresponding document id.
>    -  Time: O(count-of-k)
>
> c) Tokens in a document set D
> -  How: We iterate over each doc in D, perform an FM-Index operation to get the token position and check if it lies in the given range.
> -  Time: O(number of documents). In practise, the model is trained to generate highly specific initial keywords that result in a small number of documents.
>
> A combination of these yields all the constraints needed. We also apply a few optimizations on top of this, two of which are highlighted below:
>
> * Several computations are cached across a decoding pass, eg: position of k in index.
> * To cap the cost of computing constraints at each decoding step, we allow for unconstrained generation in rare scenarios, when the estimated cost is too high. If the generated path is absent in the corpus (<1% examples), they are awarded a 0 score.
>
>
> 4.  > “What are the run time for inference? How is it compared to other method?”
>
> Our current implementation is not optimized for inference time efficiency. Currently 1P pays a significant overhead for communication between the accelerator used for transformer decoding, and the separately stored index managed by CPU. Subsequently, the full runtime is significantly slower than well optimized retrieve-then-read methods. However, we believe this should be addressable by architectural improvements like moving the FM-index to the accelerator, implementing parallel calls to the index, caching documents.
>
> 5. > Additional Experiments
>
> To better understand whether 1P generalizes beyond the NQ task, we also tested it on WebQuestions and CuratedTrec. Both of these datasets support retrieval and answer accuracy metrics.
>
> In the below experiments, all systems, 1P, DPR and Reader are trained only on Open-NQ. We find that 1P performs better than DPR + Reader according to both the retrieval and answer accuracy metrics in this out-of-domain setting.
>
> **WebQuestions**
> ```
> 	              Hits@1  ||    EM   ||   F1
> DPR + Reader :    32.0    ||  17.3   ||  30.2
> 1P           :    38.0    ||  20.5   ||  32.7
> ```
>
> **CuratedTrec**
> ```
> 	              Hits@1  ||    EM   ||   F1
> DPR + Reader :    51.6    ||  35.0   ||   -
> 1P           :    63.8    ||  36.4   ||   -
> ```
>
> We will include these results along with associated analysis in the final version of the paper.

---

### Official Review · Reviewer_waPh · 2023-08-04

**Soundness:** 3

**Excitement:**

3: Ambivalent: It has merits (e.g., it reports state-of-the-art results, the idea is nice), but there are key weaknesses (e.g., it describes incremental work), and it can significantly benefit from another round of revision. However, I won't object to accepting it if my co-reviewers champion it.

**Paper Topic And Main Contributions:**

This paper describes 1-PAGER (or 1P), a method in which a single model both retrieves a document and generates an answer to a given question. More specifically, 1P stores documents in an FM-Index and performs constrained (to text present in the corpus) decoding to generate keywords that iteratively narrow the space of documents until a single passage is returned. The last generated keyword is treated as the model's answer. Although 1P outperforms closed QA systems, it (unsurprisingly) underperforms SOTA retrieval-augmented models which retrieve more than one document.

**Questions For The Authors:**

- Do you think planning in the form of MCTS (actual lookahead) could be helpful?
- Could you extend this method to retrieve multiple documents effectively?

**Reasons To Accept:**

+ The paper is well-written. It is very clear what the authors did and why they did it, which I certainly appreciate (as will future readers).
+ The constrained decoding method is clever and seems like an interesting direction to explore for greater factuality / grounding.
+ Desirable to have a single model that does both retrieval and answer generation.
+ Greater interpretability via keywords seems desirable as well.

**Reasons To Reject:**

- The authors clearly differentiate the work from prior methods that retrieve multiple documents (passages) and perform unconstrained generation. However, only supporting the retrieval of a single document seems like a serious limitation. It is often the case the answer is not contained in a single passage or document and can only be derived by combining information from multiple sources and potentially reasoning over that information. In this case, 1P will not be effective. 1P Unconstrained may be able to overcome this issue, but it doesn't seem particularly competitive with other approaches.
- Evaluations are performed only on a single dataset (Open-NQ).
- The generation of training data seems quite complex and full of heuristics. Will the model just learn to emulate these heuristics?
- This method is presented as desirable because it can retrieve and generate in a single pass with a single model. However, 1P+Reader outperforms 1P alone, which is somewhat disappointing / calls into question the viability of this method.
- Brittleness with respect to tokenization / exact match seems like a serious concern, though the authors have worked toward mitigating this.

**Reproducibility:**

4: Could mostly reproduce the results, but there may be some variation because of sample variance or minor variations in their interpretation of the protocol or method.

**Reviewer Confidence:**

4: Quite sure. I tried to check the important points carefully. It's unlikely, though conceivable, that I missed something that should affect my ratings.

**Typos Grammar Style And Presentation Improvements:**

The paper is overall very well-written, so there are unusually few typos / grammatical errors, which is appreciated.

"Each prefix of this search path defines subset of D via the search path corpus partitioning function." -> "a subset"
"Answer prediction is treated in exactly the same way as keyword selection and, in 1P..." -> remove comma

---

> ### Author Rebuttal · Authors · 2023-08-29
>
> Thank you for your thorough reading and valuable feedback. Below we have provided some responses to your criticisms of the current paper.
>
>
> 1. > “Evaluations are performed only on a single dataset (Open-NQ).”
>
> This is a valid criticism also raised by other reviewers. To understand this better, we tested 1P on WebQuestions and CuratedTrec. Both of these datasets, like Open-NQ, are suitable for joint retrieval + QA experiments.
>
> In the below experiments, all systems: 1P, DPR, Reader are trained only on Open-NQ. We find that 1P performs better than DPR + Reader on both the retrieval and answer accuracy metrics even in this out-of-domain setting.
>
> **WebQuestions**
> ```
> 	              Hits@1  ||    EM   ||   F1
> DPR + Reader :    32.0    ||  17.3   ||  30.2
> 1P           :    38.0    ||  20.5   ||  32.7
> ```
>
> **CuratedTrec**
> ```
> 	              Hits@1  ||    EM   ||   F1
> DPR + Reader :    51.6    ||  35.0   ||   -
> 1P           :    63.8    ||  36.4   ||   -
> ```
>
> We will include these results along with associated analysis in the final version of the paper.
>
>
> 2. > “The authors clearly differentiate the work from prior methods that retrieve multiple documents (passages) and perform unconstrained generation. However, only supporting the retrieval of a single document seems like a serious limitation. It is often the case the answer is not contained in a single passage or document and can only be derived by combining information from multiple sources and potentially reasoning over that information. In this case, 1P will not be effective.”
>
> We agree that question answering can often require multi document reasoning but also point out that most datasets commonly used for open-domain QA generally have the presence of an answer in a single supporting document. This is a prerequisite for answers in NQ, in particular.
>
> When reasoning over multiple retrievals for NQ and associated datasets, the standard retrieve-then-read methods do not generally perform multi-hop reasoning but are instead relying on the reader to identify which of 100 retrievals are most relevant, similar to re-rankers used in the IR literature. For this reason, we feel it is reasonable to compare 1P to top-1 retrievals to get a clear picture of the model's performance at the pure retrieval task. Note that 1P is capable of retrieving multiple documents as depicted in Appendix A7.
>
> We have not addressed the more challenging task of multi-document reasoning but believe this could be a good next area of focus for 1P. Pipelined systems need to chain together multiple steps of document retrieval and document reading for true multi-hop tasks. By integrating retrieval into the decoding process directly, 1P could simplify this process and support better end-to-end learning of multi-hop tasks.
>
>
>
> 3. > “The generation of training data seems quite complex and full of heuristics. Will the model just learn to emulate these heuristics?”
>
>  The heuristics used to kick-start training are not elegant but we have evidence that 1P is not simply emulating these at test time.
>
> * The heuristics themselves are not sufficient---1P also needs LM scoring to find good search paths. (Table 5)
>
> * Our experiments with Unconstrained-1P show that simply replicating the training distribution is insufficient, the constraints introduced in 1P are essential to ensuring that the model can navigate the evidence corpus effectively.
>
> * The qualitative analysis presented in Table 4 also illustrates diverse patterns of search paths generated in different scenarios that are more effective and context dependent than the heuristically generated paths used during training.
>
> * Finally, the out of domain results on WebQuestions and CuratedTrec presented above show that the choice of  training-time heuristics does not hurt generalization to new tasks.
>
> Future work should explore even more general methods of training.
>
>
>
> 4. > “However, 1P+Reader outperforms 1P alone, which is somewhat disappointing / calls into question the viability of this method.”
>
> In this paper, we aim to explore the spectrum between CBQA, where the answer is generated without reference to an evidence corpus, and pipelined approaches that feed retrieved documents into the transformer.
>
> We agree that the pipelined approach is still most performant and do not yet argue for 1P as the best solution for anyone aiming to build the most accurate QA system possible. However, we believe that 1P's improvement over the equivalent CBQA model is significant and we also believe that 1P adds value in demonstrating how a single transformer model can be harnessed to do both retrieval and  answering, which has not been done before.
>
>
> 5. > “Do you think planning in the form of MCTS (actual lookahead) could be helpful?”
>
> Thank you for the suggestion. We agree that better search algorithms would be extremely beneficial. While we have initially focused on a proof-of-concept with the beam search algorithm that is commonly used for Transformer decoding we agree that MCTS would likely be a better fit to help 1P recover from poor choices. We refer to Table 2 for an initial investigation of how beam search improves 1Ps performance over greedy decoding.
>
>
> 6. > “Brittleness with respect to tokenization / exact match”
>
> We acknowledge that 1P is sensitive to exact matches. This is a general limitation with lexicalized retrieval but is countered by the model's ability to directly match rare named entities (e.g. EMNLP 2023) and generate interpretable search paths. We believe that there is value in exploring lexicalized approaches that can both generalize over variations in surface form, via paraphrasing, and also capture highly discriminative n-grams. However, we also agree that there is more to be done to ensure that 1P is more robust to minor variations in tokenization.

---

### Official Review · Reviewer_RLpb · 2023-08-04

**Soundness:** 4

**Excitement:**

4: Strong: This paper deepens the understanding of some phenomenon or lowers the barriers to an existing research direction.

**Paper Topic And Main Contributions:**

This paper introduces an approach to handle open-domain QA problems using generative language models, i.e., the 1-pager approach. Instead of using tf-idf/BM25/dense retrieval, the authors propose to use a single generative language model to generate both the keywords of the relevant document and the final answer in only one forward inference process. The authors also make sure the generated keywords of the relevant documents indeed appear in some documents of the knowledge base by introducing constraints in the decoding process of the generative language model.

The proposed approached is realized with a T5 model, evaluated on Open-NQ dataset. Results show that the proposed approach yields competitive performance to the SOTA retrieve-and-read approach (such as DPR + T5 reader). A few other experiments are conducted to investigate a few different variations of the proposed approach.

Although the generation-with-constraint method is very similar to SEAL, 1-pager uses different heuristics to generate the keywords of the relevant documents. Experiments show that these modifications lead to some improvements to the retrieval performance compared with SEAL.

**Reasons To Accept:**

- Although the idea of using generative language model to generate the keywords of documents is explored in SEAL, the authors of 1-pager change the way the keywords are generated by scoring the keywords in the training data. Ablation experiments show that the scoring of keywords during training is important to the later keyword generation.
 - The paper is well presented and easy to follow. Most of the details are included.
 - 1-pager yields pretty good results on the Open-NQ dataset that are competitive to DPR.

**Reasons To Reject:**

- 1-pager is only evaluated on Open-NQ. It is not clear whether the improvement of retrieval performance is associated with some particular artifacts of the dataset.
 - Although one of the selling point of 1-pager is to generate the document keyword and answer at the same time, which is different from the retrieve-and-read paradigm, it seems to me that 1-pager with the reader makes more sense. As discussed in the paper, if 1-pager does not use retrieve-and-read, the answer generation will be only conditioned on the generated keywords, instead of the retrieved document. This will lose some important information in the answer generation process. Therefore 1-pager with reader yields better performance than the no reader version of 1-pager.

**Reproducibility:**

4: Could mostly reproduce the results, but there may be some variation because of sample variance or minor variations in their interpretation of the protocol or method.

**Reviewer Confidence:**

4: Quite sure. I tried to check the important points carefully. It's unlikely, though conceivable, that I missed something that should affect my ratings.

---

> ### Author Rebuttal · Authors · 2023-08-29
>
> Thank you for your careful reading of our paper and valuable feedback. Below we have provided some responses to your two criticisms of the current paper.
>
> 1.  > “1-pager is only evaluated on Open-NQ”
>
> This is a valid criticism also raised by other reviewers. To understand this better, we also tested 1P on WebQuestions and CuratedTrec. Both of these datasets, like Open-NQ, are suitable for joint retrieval + QA experiments.
>
> In the below experiments, all systems: 1P, DPR, Reader are trained only on Open-NQ. We find that 1P performs better than DPR + Reader on both the retrieval and answer accuracy metrics even in this out-of-domain setting.
>
> **WebQuestions**
> ```
> 	              Hits@1  ||    EM   ||   F1
> DPR + Reader :    32.0    ||  17.3   ||  30.2
> 1P           :    38.0    ||  20.5   ||  32.7
> ```
>
> **CuratedTrec**
> ```
> 	              Hits@1  ||    EM   ||   F1
> DPR + Reader :    51.6    ||  35.0   ||   -
> 1P           :    63.8    ||  36.4   ||   -
> ```
>
> We will include these results along with associated analysis in the final version of the paper.
>
> 2. > “ Although one of the selling point of 1-pager is to generate the document keyword and answer at the same time, which is different from the retrieve-and-read paradigm, it seems to me that 1-pager with the reader makes more sense. As discussed in the paper, if 1-pager does not use retrieve-and-read, the answer generation will be only conditioned on the generated keywords, instead of the retrieved document. This will lose some important information in the answer generation process. Therefore 1-pager with reader yields better performance than the no reader version of 1-pager.”
>
> In this paper, we aim to explore the spectrum between CBQA, where the answer is generated without reference to an evidence corpus, and pipelined approaches that feed retrieved documents into the transformer.
>
> We agree that the pipelined approach is still most performant and do not yet argue for 1P as the best solution for anyone aiming to build the most accurate QA system possible. However, we believe that 1P's improvement over the equivalent CBQA model is significant and adds value in demonstrating how a single transformer model can be harnessed to do both retrieval and  answering, which has not been done before.

---

### Meta-Review · Area_Chair_BLKU · 2023-09-21

**Recommendation:** 4

**Metareview:**

The paper proposes using a single LLM to, given a query, generate keywords for retrieval and then produce the final answer in one pass. Constrained decoding is adeptly implemented, using the FM-index, to ensure that the documents contain the specified keywords. Experiments validate the efficacy of the proposed method on Open-NQ (as presented in the paper), as well as on WebQuestions and CuratedTrec in a zero-shot manner, as recommended by the reviewers (details reported in the rebuttal). The paper is excellently presented with clear motivation. We strongly urge the authors to address and refine the areas pinpointed by the three reviewers.

---

### Decision · Program_Chairs · 2023-10-07

**Decision:**

Accept-Findings

**Comment:**

The paper proposes using a single LLM to, given a query, generate keywords for retrieval and then produce the final answer in one pass. Constrained decoding is adeptly implemented, using the FM-index, to ensure that the documents contain the specified keywords. Experiments validate the efficacy of the proposed method on Open-NQ (as presented in the paper), as well as on WebQuestions and CuratedTrec in a zero-shot manner, as recommended by the reviewers (details reported in the rebuttal). The paper is excellently presented with clear motivation. We strongly urge the authors to address and refine the areas pinpointed by the three reviewers.